# Quantum sensing with spin defects in boron nitride nanotubes

Roberto Rizzato [1,2] ✉, Andrea Alberdi Hidalgo[1], Linyan Nie[1,2], Elena Blundo [2,3], Nick R. von Grafenstein [1,2], Jonathan J. Finley [2,3] & Dominik B. Bucher [1,2] ✉

Spin defects in semiconductors are widely investigated for various applications in quantum sensing. Conventional host materials such as diamond and hexagonal boron nitride (hBN) provide bulk or low-dimensional platforms for optically addressable spin systems, but often lack the structural properties needed for chemical sensing. Here, we introduce a new class of quantum sensors based on naturally occurring spin defects in boron nitride nanotubes (BNNTs), which combine high surface area with omnidirectional spin control, key features for enhanced sensing performance. First, we present strong evidence that these defects consist of weakly-coupled spin pairs, akin to recently identified centers in hBN, and demonstrate coherent spin control over ensembles embedded within randomly oriented, dense, BNNTs networks. Using dynamical decoupling, we enhance spin coherence times by a factor exceeding 300 times and implement high-resolution detection of radio-frequency signals. By integrating the BNNT mesh sensor into a microfluidic platform we demonstrate chemical sensing of paramagnetic ions in solution, with detectable concentrations reaching levels nearly 1000 times lower than previously demonstrated using comparable hBN-based systems. This highly porous and flexible architecture positions BNNTs as a powerful new host material for quantum sensing.

Optically active spin defects in solid-state systems have emerged as fundamental tools in quantum technology and sensing[1–3]. The nitrogen-vacancy (NV) center in diamond[4,5] (Fig. 1a) has become a well-established platform, enabling proof-of-concept breakthroughs including single-spin and single-molecule detection[6–14], high-resolution nuclear magnetic resonance (NMR)[15–18], and nanoscale sensing of chemical interactions at the liquid-solid interface[19–24]. Additionally, extensive research has focused on leveraging the high, optically pumped electronic spin polarization of these systems to hyperpolarize nuclear spins[25–30], thereby enhancing signal strength in NMR spectroscopy[16,31]. Despite these advancements, it has become evident that the success of both sensing and hyperpolarization critically depends on maximizing the

sensor–target interaction, a task that remains challenging for state-of-the-art solid-state systems[23,32,33]. Hexagonal boron nitride (hBN), a well-known two-dimensional (2D) van der Waals material, has recently emerged as a promising alternative[34–39] due to the possibility of incorporating stable spin defects[34,40–44] into ultra-thin hBN flakes[45]. This enables the sensors to be positioned in proximity to the sample, with the potential to embed these nanoscale 2D sensors directly into target systems, e.g., layered heterostructures[40,44,46–54]. While this approach holds great promise for solid-state and materials research, it remains less suited for soft-matter environments and chemical sensing, where detecting molecular processes requires intimate interaction and often multiple contacts between spin defects and analytes[55–59].

---

[1]Technical University of Munich, TUM School of Natural Sciences, Chemistry Department, Lichtenbergstraße 4, Garching bei München, München, Germany. [2]Munich Center for Quantum Science and Technology (MCQST), Schellingstr. 4, München, Germany. [3]Walter Schottky Institute, TUM School of Natural Sciences, Physics Department, Am Coulombwall 4, Garching bei München, München, Germany. ✉e-mail: roberto.rizzato@tum.de; dominik.bucher@tum.de

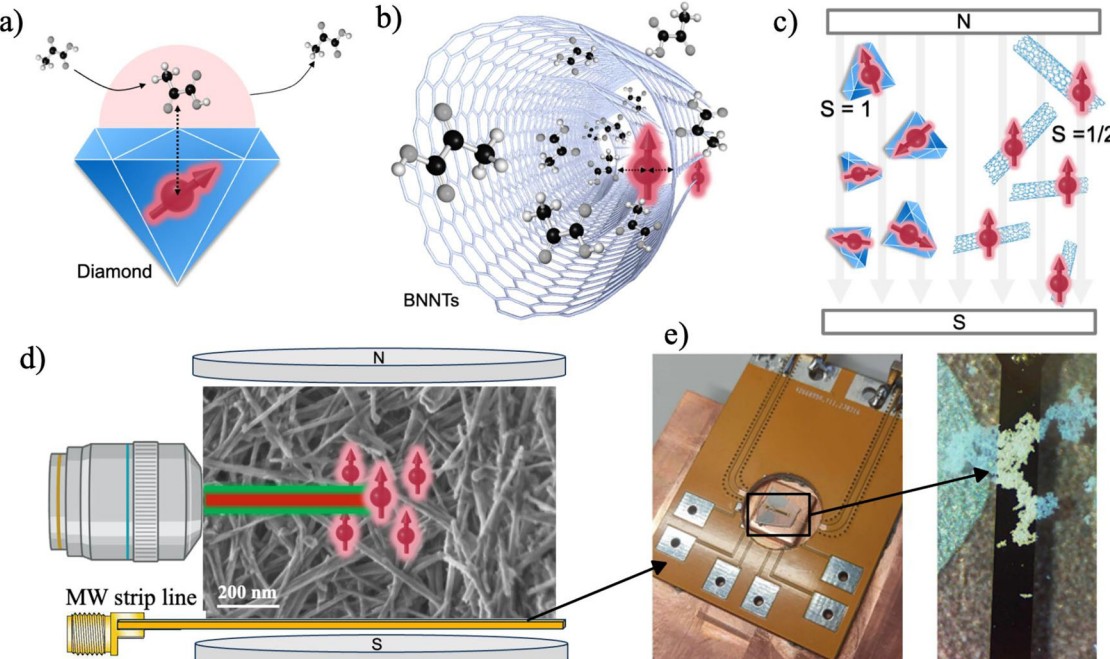

**Fig. 1 | Conventional solid-state spin defects vs. novel spin-pairs in BNNTs.**
**a** Schematic of an NV center in diamond, where the analyte sits on top, and the spin−target distance is set by the NV depth (e.g., 5–10 nm); analyte confinement at the spin sensor is challenging. **b** In BNNTs, the defects are in the hBN walls, potentially providing closer proximity, and the sample can go inside the nanotube, providing natural confinement in a wrap-around geometry. When taken as ensembles, BNNTs form a mesh architecture characterized by a nanoporous, high-surface-area material suitable for chemical sensing and spin-hyperpolarization applications. **c** NV centers are $S = 1$ systems with a zero-field splitting; in randomly oriented nanodiamonds, the NV quantization axs are also randomly oriented, and

consequently, only a small fraction of the defects are aligned to the bias field and are active for quantum sensing. By contrast, BNNT spin-pair defects behave as $S = \frac{1}{2}$ systems, aligning directly with the external bias field and thus providing a well-defined quantization axis regardless of the nanotube orientation. **d** SEM image of our BNNT mesh sample positioned on a microwave (MW) strip line for spin control and optically detected magnetic resonance (ODMR) characterization. Optical excitation is provided by laser illumination, and defect photoluminescence is collected through an objective. **e** Photographs of the MW delivery setup, with a close-up view of the gold strip line where the BNNT mesh is placed.

In this work, we introduce spin defects in boron nitride nanotubes (BNNTs)[60,61] (Fig. 1b) as a quantum sensing platform for chemical analysis. In particular, we exploit two key features:

1) The intrinsic nanostructure of BNNTs enables mesh-like architectures, comprising nanotubes with hollow interiors and accessible surfaces[62–65], and offering key advantages for applications requiring molecular confinement or high spin defect-to-target ratios (Fig. 1b)[26,55–57,66–68].

2) Unlike NV centers in diamond or $V_B^-$ centers in hBN and in BNNTs[61], which require a well-aligned bias magnetic field for qubit selection due to their preferential quantization axis[26,40,61,67–70], the spin defects investigated in our BNNTs are optically active $S = \frac{1}{2}$ systems which therefore show an isotropic magnetic response, allowing for omnidirectional spin control[41–43,60,71–74] (see Fig. 1c).

This is a key property for the development of a novel high-surface-area sensing platform, which can now be composed of randomly oriented nanoparticles (or nanotubes, see Fig. 1d), where all embedded spin defects in BNNTs remain equally active regardless of the orientation.

We first characterize the properties of the spin ensembles and provide strong indications that they may correspond to the weakly-coupled spin-pair defects (e.g., carbon-related $C_?$-defects) recently identified in hBN crystals[71,73,74]. Then, we employ dynamical decoupling protocols to extend their room-temperature coherence times and implement state-of-the-art quantum sensing methods to achieve high-resolution radiowave frequency (RF) signal detection. Finally, we demonstrate the practical potential of this platform by integrating the BNNT-based mesh sensor into microfluidics. By monitoring the effect

of paramagnetic ions on the spin properties of the defects, we successfully detect $Gd^{3+}$ ions at micromolar concentrations−approximately three orders of magnitude more sensitive than what has been reported using spin defects in hBN nanosheets[75,76]. These results highlight the versatility and promise of BNNT-based quantum sensing for applications in future chemical detection and hyperpolarization techniques.

## Results and discussion
### Characterization of spin-defects in BNNTs
All experiments throughout this work were performed under ambient conditions. We utilized multiwalled BNNTs as received without any additional treatments, and directly drop-cast on a microwave (MW) strip line (see Fig. 1e and Sample Preparation in the "Methods" section for more details). Before discussing the spin properties, we first measured the photoluminescence (PL) spectrum of the emitting defects naturally present in BNNT ensembles (Fig. 2a). The spectrum exhibits a broad emission band reminiscent of those reported for carbon-related spin defects in boron nitride materials[60,71,74,77], indicating that the observed centers may belong to this defect family. We then characterized the spin defects using optically detected magnetic resonance (ODMR). Figure 2b presents ODMR spectra recorded at varying magnetic fields, revealing a single resonance line that shifts with the applied magnetic field and reaches a maximum contrast of 0.3−0.4% after optimizing MW and laser powers and pulse durations. Notably, the resonance remained identical and featureless regardless of the magnetic field orientation, consistent with the isotropic Zeeman interaction of spin-½ systems, whose quantization axis aligns with the

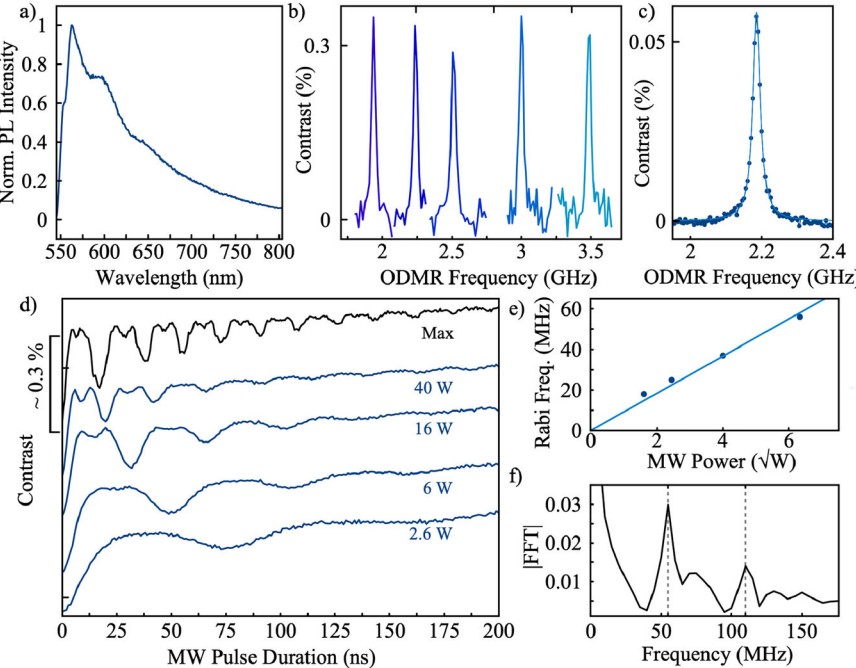

**Fig. 2 | Characterization of spin defects in an ensemble of BNNTs. a** Photoluminescence (PL) spectrum of the emitting defects. **b** ODMR spectra recorded at different bias magnetic field strengths. **c** ODMR spectrum of BNNT spin defect ensembles at 78 mT. Blue dots show the experimental data, with a fitted Lorentzian line shape overlaid in light blue. **d** Rabi oscillations recorded at various input MW powers (in Watts). The black line shows the experimental Rabi oscillation under optimized conditions of maximum input MW power, showing a pronounced beating pattern. **e** Rabi frequency plotted as a function of the square root of the MW power, confirming the expected linear dependence. **f** Fast Fourier transform (FFT) of the Rabi oscillation at max. MW power revealing two main frequency components: a dominant peak near 50 MHz and a smaller peak at twice that frequency, typically observed in spin pairs.

external field rather than with any crystallographic direction. Figure 2c shows the ODMR spectrum acquired using minimal laser and MW powers to reduce spectral broadening. The resonance is well fitted by a single Lorentzian with a full width at half maximum of ~25 MHz. This linewidth is consistent with that reported for single-spin defects in BNNTs[60] and about twice as narrow as that predicted for spin-pair complexes in hBN[74]. No significant inhomogeneous broadening—typically observed in ensembles—was detected, nor was hyperfine structure, suggesting that couplings to nearby nuclear spins (e.g., $^{11}B$ or $^{14}N$) are either weak or masked by the ~ 25 MHz linewidth.

We continue to demonstrate that we can efficiently control ensembles of spin defects in BNNTs under ambient conditions. This is achieved through Rabi oscillation shown in Fig. 2d, recorded by driving the spins at the resonance frequency and varying the MW pulse duration from 0 to 200 ns. These measurements were performed using different MW powers, revealing the expected scaling of Rabi frequencies with the square root of the MW power (Fig. 2e). Notably, at increased MW power levels, a distinct beating pattern emerges in the Fourier transform of the Rabi trace revealing two distinct peaks: a dominant component and a smaller one at twice the frequency (Fig. 2f). A similar effect was recently observed by Scholten et al. for $C_2$-defects in hBN crystals, where it was attributed to weakly coupled spin-½ pairs[71]. The presence of this beating pattern, along with the optical spectrum observed in our BNNTs (Fig. 2a), further suggests that we observe a similar defect type. Furthermore, the positive contrast in the ODMR and its magnitude align well with *Optical-Spin Defect Pair* (OSDP) model recently proposed by Robertson et al.[74] for ensembles of $C_2$-defects in hBN crystals. According to the model, these spin defects may arise from electron hopping between two nearby defect sites, where optical activity occurs when both electrons are localized in a singlet state at the same site. Upon hopping, the electrons can form a weakly coupled radical pair, giving rise to both spin-parallel and spin-antiparallel states. When subjected to a bias magnetic field, these states are split by the Zeeman interaction and can be addressed by MW radiation.

## Spin relaxation and coherence extension

After obtaining strong indications of the presence of spin-pair defects in BNNTs and demonstrating the coherent manipulation of the randomly oriented ensemble, we continue to explore advanced spin control. Firstly, we start with the measurement of the $T_1$ spin-lattice relaxation time (Fig. 3a). We monitor the fluorescence contrast as the time interval T between two laser readouts increases. By fitting the decay, we extract a $T_1$ relaxation constant of approximately 26 μs. According to Robertson et al.[74] we observe that the measured fluorescence increases with time T, which may be due to the recovery of the ground state population, after optical initialization (see inset of Fig. 3a and "Methods" section for more details). Consequently, the $T_1$ relaxation time may not reflect the pure spin–lattice relaxation. Therefore, while we use the term $T_1$ for consistency with prior literature, we emphasize that it should not be directly interpreted as a pure spin-lattice relaxation time, nor compared quantitatively with $T_1$ values obtained in different platforms or under different measurement conditions.

We then measured the native coherence time $T_2$ using a spin-echo sequence (Fig. 3b). In this experiment, after initializing the spins with a laser pulse, a microwave π/2-pulse resonant with the defects' ODMR line creates spin coherence, which is refocused by a single π-pulse before being transformed back into spin populations by a final π/2-pulse. By monitoring the detected signal while varying the inter-pulse delay and, consequently, the total sequence time T, we recorded an exponential decay. Fitting this decay yields a $T_2$ time constant of approximately 50 ns, which is about half the value reported for single defects in individual BNNTs[60].

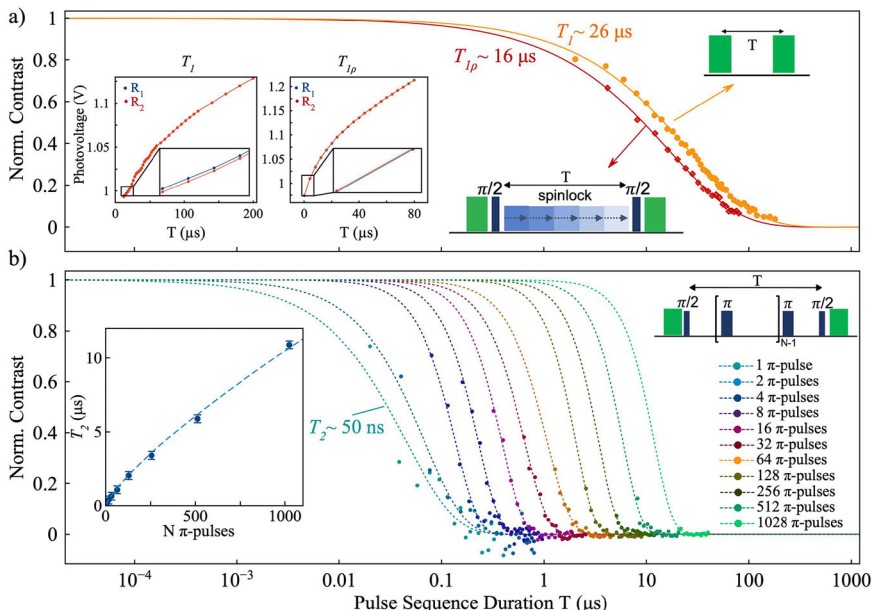

**Fig. 3 | Spin Relaxation and coherence extension of an ensemble of spin defects in BNNTs.** Semi-log plot of spin coherence and relaxation decays measured using various pulse sequences, illustrated in the insets. Colored dots represent experimental data, while lines correspond to fits. **a** The solid orange data points and fit depict the fluorescence contrast decay for the $T_1$ spin-lattice relaxation measurement (sequence shown in the top-right inset). The solid red squares and line correspond to the coherence decay measured via spin-locking experiments, yielding $T_{1\rho}$ (pulse sequence shown in the inset on the left). In the inset, the raw data corresponding to both experiments are shown before normalization. $R_1$ and $R_2$ represent the two readouts for measurement and normalization/noise subtraction, respectively (see "Methods" section for more details). **b** The cyan dashed curve and dots show coherence decay measured using a simple Hahn-echo sequence (one π-pulse), from which the $T_2$ constant is extracted. All other dashed curves and colored dots represent coherence decays measured using CPMG sequences with an increasing number of π-pulses, $N$ (pulse sequence shown top-right inset). The bottom-left inset plots the coherence times $T_{2,\,CPMG}^{(N)}$ as a function of $N$, fitted to a power law model.

Furthermore, we present data on the extension of the coherence times via dynamical decoupling methods. In Fig. 3b, data from multiple Carr-Purcell-Meiboom-Gill (CPMG) experiments are shown (colored dots and dotted lines). CPMG is a spin-echo-based pulse scheme[1,53,78,79], that uses a train of π-pulses for decoupling the spins from the environmental magnetic noise. This process effectively prolongs the $T_2$ of the BNNT spin defects, and we can determine the corresponding increase in $T_{2,\,CPMG}^{(N)}$, with the number N of π-pulses. Remarkably, residual coherence is still observed even after ~$10^3$ π-pulses, where we reach coherence times of over 10 μs. We plot the coherence time as a function of the number of π-pulses $N$ in the inset in Fig. 3b. It follows a sub-linear trend, which we fit with a power-law $T_2 = aN^s$, yielding an exponent $s \approx 0.79$. The extracted exponent is slightly higher than the theoretical scaling $T_2 \propto N^{2/3}$ expected for a Lorentzian spin bath in the slow-noise limit ($\tau_c \gg T$). Alongside the featureless ODMR spectrum—indicating weak coupling with nearby spins—this suggests deviations from an ideal Lorentzian model, possibly due to reduced bath spin density and more correlated (non-Markovian) dynamics[53,80–82].

Finally, we demonstrate coherence extension using the spinlock sequence[25,53,83], in Fig. 3a. In this method, after generating a superposition with a π/2 pulse, the spin is locked onto the axis by applying a longer MW pulse with a phase aligned with the spin vector. By monitoring the fluorescence contrast while sweeping the spin-lock pulse duration, we can measure the relaxation time in the rotating frame ($T_{1\rho}$), which is significantly longer than the native $T_2$ time, extending from ~50 ns up to ~16 μs and approaching the $T_1$ limit. Similarly to what we have observed for the $T_1$ measurement before, we acknowledge that the extracted $T_{1\rho}$ and $T_2$ values may not directly represent pure spin coherence times, especially in regimes where the decay approaches the measured $T_1$. Consequently, these values should be interpreted cautiously and not assumed to correspond exactly to intrinsic spin relaxation times. Nevertheless, this remarkable over-300-fold

increase in coherence time highlights the effectiveness of the dynamical decoupling protocols in mitigating decoherence mechanisms of our randomly oriented BNNT ensemble. Notably, the intricate Rabi dynamics of weakly coupled spin-½ pair ensembles posed challenges for precise microwave pulse calibration, highlighting the need for more systematic studies in future work. A description of the optimization procedure used in this study is provided in the "Methods" section.

## Sensing of RF signals

Having demonstrated the effectiveness of dynamical decoupling protocols in enhancing coherence, we use our BNNTs spin defect ensemble to detect RF signals generated from an antenna placed nearby (Fig. 4a). This is made possible by leveraging spin coherence, which enables the encoding of a relative phase in response to an external magnetic field. In the presence of oscillating magnetic fields, such as RF signals, this phase can be coherently accumulated using spin-echo-based techniques. By precisely tuning the timing of the pulse sequence (i.e., the delays between π-pulses) to match the oscillation period of the RF field, constructive phase buildup is achieved. This *tuning* allows the sensor's spin coherence to constructively integrate the field's effect, enabling high-sensitivity RF detection[1,53]. We used the Coherently Averaged Synchronized Readout (CASR)[15,53,84], which phase-locks a series of concatenated spin-echo sequences to the target RF field (Fig. 4b). Each pulse sequence is tuned to the RF frequency and designed so that its total duration exactly matches an integer multiple of the RF wave's period. When fully matched, each sequence experiences the same phase of the RF field in every cycle, resulting in an identical fluorescence readout. However, when slightly detuned, the protocol effectively converts the RF signal into a low-frequency optical beating over time that encodes spectral information. By continuously collecting data over an extended period, CASR achieves high frequency resolution, as this is limited not by the

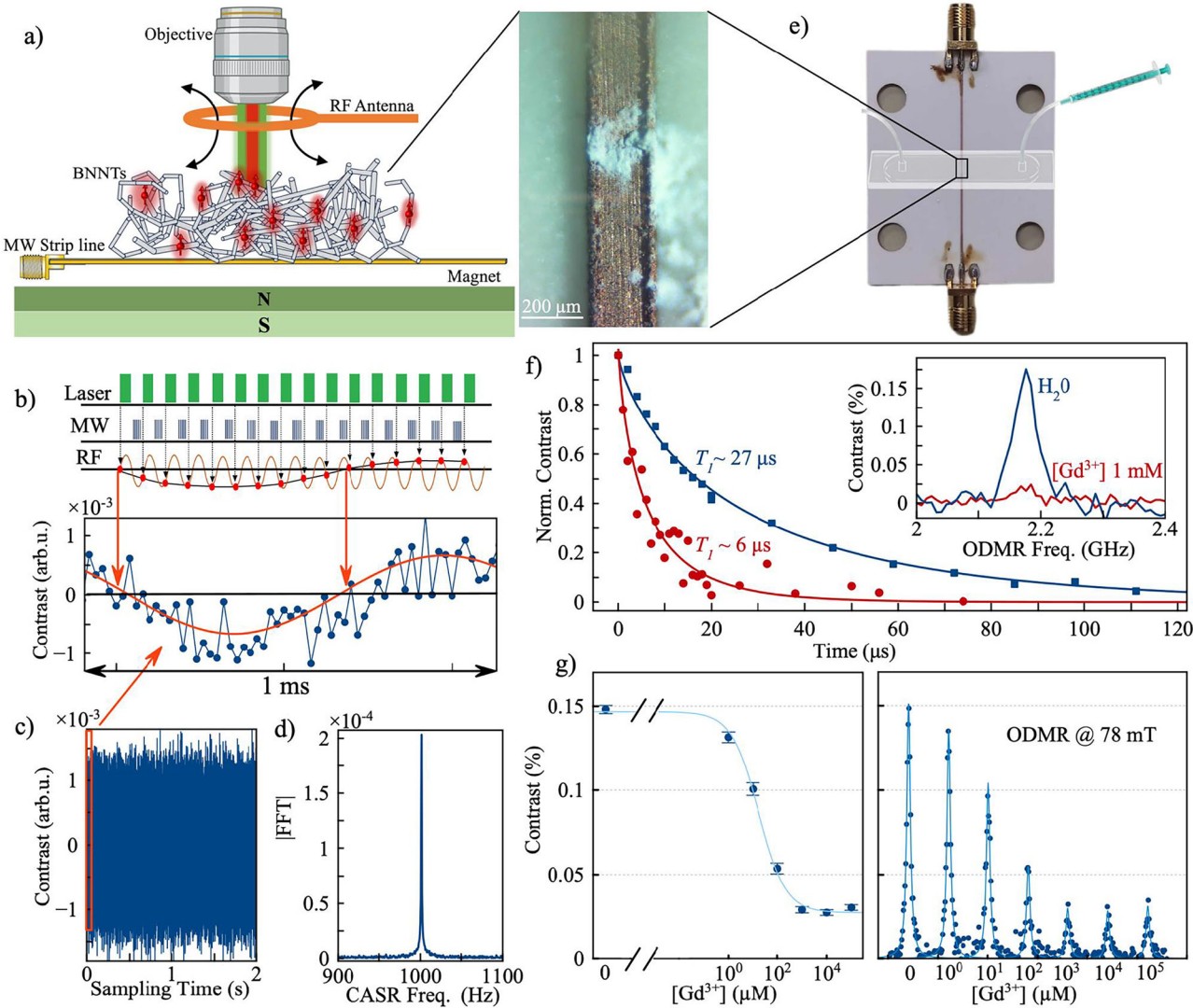

**Fig. 4 | Quantum sensing with ensembles of spin defects in BNNTs. a** Simplified schematic of our setup for the detection of RF signals through the BNNTs mesh sensor. **b** Top: Coherently Averaged Synchronized Readout (CASR) pulse sequence. 2-$\pi$ pulses-spin echo subsequences are synchronized to the RF sensing frequency for 2 s. Each detected point corresponds to an optical readout, represented by the red circles. Bottom: Zoom-in of the experimental CASR time. **c** CASR time trace. **d** Fourier transformation of the time trace in (**c**) reveals a sharp peak at the CASR frequency. **e** Integrated microfluidic chip for in-solution quantum sensing using a BNNTs mesh. A copper MW strip line enables coherent spin control, while the microfluidic chip−open at its bottom−is directly mounted on top of the strip line and sealed. Inlet and outlet tubing enable solution flow across the sensing region. The zoom-in highlights the area where BNNTs are deposited onto the strip line,

appearing as a porous, textured white powder. **f** $T_1$ relaxation curves of spin defects in BNNTs exposed to distilled water (blue) and to a 1 mM solution of $Gd^{3+}$ ions (red), along with corresponding exponential fits. The presence of paramagnetic ions results in a pronounced reduction in spin relaxation time. Inset: Corresponding ODMR spectra under both conditions. **g** $Gd^{3+}$ concentration dependence of the ODMR contrast. Left: Semilog plot of the maximum ODMR contrast as a function of $Gd^{3+}$ concentration. The data are well fit by a Hill-like function (light blue), indicating a progressive and saturable decrease in contrast. Right: Representative ODMR spectra acquired at $B_0 = 78$ mT, corresponding to the data points shown on the left, illustrating the progressive reduction in signal intensity with increasing $Gd^{3+}$ concentration.

spin coherence time, but by the timing stability of the control electronics. To demonstrate the implementation of this method for our randomly oriented BNNTs, we applied a 15 MHz RF signal and introduced a deliberate 1 kHz detuning for a total detection time of 2 s. We present the Fourier transform of the detected time-dependent signal in Fig. 4b, c, yielding a sharp peak with a full-width at half-maximum of ~1 Hz, as shown in Fig. 4d. Additionally, we employed the same CASR protocol to evaluate the achievable sensitivity of the mesh sensors, obtaining a sensitivity of approximately 20 μT/√(Hz). This value is approximately ten times higher than that demonstrated using ensembles of $V_B^-$ centers in hBN[53] and therefore does not yet match the performance of more established quantum sensing platforms. However, it is important to emphasize that this system is still in its infancy.

The current results were achieved using naturally occurring spin defects in as-received BNNT material, without any form of defect engineering or material optimization. Importantly, the ability to apply advanced quantum control protocols−such as NOVEL[27,85] or PulsePol[30,67]−to spin defects in a randomly oriented host paves the way for implementing solid-state, spin-based hyperpolarization techniques.

## Sensing of paramagnetic ions
Spin-defect-based quantum sensors offer a powerful platform for detecting paramagnetic ions at the nanoscale, with key applications in chemical, biological, and medical sciences[23,70,76,86–90]. This functionality is enabled by the sensitivity of the spin defect's $T_1$ to magnetic noise

generated by paramagnetic species in the nearby environment. Building on this principle, we demonstrate the first practical realization that fully embodies the original vision behind our BNNT mesh sensor, combining high surface area[91] and omnidirectional spin response with straightforward integration into a microfluidic sample-handling system[92]. To achieve this, we designed the configuration shown in Fig. 4e, where a microfluidic channel with an open bottom was placed on top of the MW strip line, where the BNNTs were drop cast (see "Methods"), ensuring direct contact between the liquid sample and the mesh sensor. A solution with varying concentrations of $Gd^{3+}$ ions was introduced, and their presence was successfully detected by monitoring changes in the spin properties of the BNNT-based spin defects (Fig. 4f). When in contact with pure distilled water, the defects exhibit a $T_1$ time of approximately $27\,\mu s$, consistent with the value observed in our previous experiments on BNNT samples in air. Upon exposure to a $1\,mM$ solution of $Gd^{3+}$ ions, the $T_1$ time significantly decreases, along with a reduction in ODMR contrast (Fig. 4f, inset). The direct correlation between the two experiments arises because both the optical initialization/readout and the microwave pulses in our ODMR protocol last $5\,\mu s$ each, resulting in a total sequence duration that is on the order of the $T_1$ timescale. When $T_1$ shortens, as in the presence of $Gd^{3+}$ ions, spin populations relax more rapidly during the measurement, reducing the spin-state contrast and thus the ODMR signal[76]. This makes ODMR contrast an effective and convenient proxy for detecting changes in relaxation dynamics. The BNNT mesh sensor can be regenerated by flushing the system with an acidic solution (pH $\sim 2$), followed by rinsing with pure water, which fully restores its performance. Finally, we examined the impact of paramagnetic $Gd^{3+}$ ions on the ODMR contrast (Fig. 4g), revealing a cooperative-binding-like interaction where local magnetic noise from $Gd^{3+}$ ions induces saturable contrast quenching (fit in Fig. 4g). The fit yields a constant of $K_d \approx 17\,\mu M$, indicating that half of the maximal quenching occurs at this concentration. Furthermore, the nonzero baseline suggests that a subpopulation of spin defects remains unaffected, likely due to their buried location within the BNNT nanotube (see "Methods" for details). While a direct comparison with previously reported systems is challenging—due to variations in structural architectures, defect types, and experimental conditions—our results underscore the distinct advantages of the BNNT mesh sensor, which enables detection of paramagnetic ions in the low micromolar range. This detection capability exceeds by several orders of magnitude what has recently been demonstrated with $V_B^-$ defects in hBN-based platforms[75,76]. Remarkably, this enhancement is achieved despite those systems exhibiting superior intrinsic spin properties—such as up to -100 × higher ODMR contrast and comparable relaxation times. Our working hypothesis is that the improved performance of our BNNT sensor is driven by its unique architecture and high surface accessibility, rather than the intrinsic qualities of the spin defects themselves. Unlike flat hBN flakes, the BNNT mesh forms a nanoporous, high-surface-area network that may enable local confinement of analytes in close proximity to spin-active regions. The nanotubes are hollow and open at both ends, allowing analytes to diffuse into the inner cavity and interact with spin defects within the tube walls. In addition, the cylindrical geometry provides radial access, in contrast to the one-sided exposure in flat 2D materials and may facilitate local concentration effects through analyte accumulation within the mesh. These findings highlight the promise of BNNT-based quantum sensors for chemical sensing applications.

## Potential applications of spin-pair defects in BNNTs

Spin-pair defects in BNNTs have the potential to realize opportunities envisioned for more established solid-state spin qubits in diamond, which were so far limited by practical constraints. This potential arises due to the unique combination of the following features: (1) a layered hBN structure that allows defects to be embedded within only a few atomic layers, enabling close proximity to target systems[45]; (2) a hollow nanotube geometry that naturally provides molecular confinement and facilitates the accumulation or the trapping of analytes at the sensor site[62,64,65,93]; (3) an isotropic magnetic response (spin $S = \frac{1}{2}$) that removes the need for precise alignment with external magnetic fields[60], making them effective in dynamic environments or in disordered configurations such as the high-surface-area mesh demonstrated in this work. Based on this, we envision several major applications:

Starting from the nanoscale, BNNTs can operate as spin sensors in dynamic or disordered environments where—unlike NV centers in nanodiamonds—they could, in principle, support the full range of quantum sensing protocols due to the lack of an intrinsic quantization axis. This could lead to the development of nanoscale NMR sensors that can be introduced into biological systems. Similarly, the BNNTs could be applied as integrated in-situ or in-operando sensors[94] in energy-conversion technologies such as batteries, fuel cells, and electrocatalytic interfaces. Furthermore, the unique structure of the BNNTs can open new opportunities in the fields of nanofluidic sensing[95,96] and nanopore technologies[97]. When integrated into nanopore platforms, spin-pair defects in BNNTs could provide local quantum sensing capabilities—for example, detecting ionic current signatures or probing trapped molecules[64].

In addition, BNNT structures are highly scalable: they can be synthesized with diameters ranging from just a few nanometers to several hundred nanometers[62], and can be deployed either as individual nanotubes[60] or as dense arrays of vertically aligned tubes[98] or disordered meshes extending up to macroscopic dimensions. Comparable approaches could, in principle, also be realized using spin-½ defects in hBN[41–43,71–74,77], either by engineering highly controlled nanochannels[99–101]—which require nanofabrication and complex sample preparation—or by employing hBN nanopowders[75,102]. However, BNNTs may offer a more attractive alternative, as they demand no additional processing and intrinsically provide a tubular geometry with hollow interiors and radially accessible surfaces. When assembled into porous meshes, this scalable architecture naturally forms nanocapillary channels that enhance wettability and molecular confinement[64,103–105].

Shifting toward larger scales, spin defects in BNNTs can find applications in environmental chemistry for water treatment[106], where, in addition to the filtration function already offered by the BNNTs meshes, the integration of quantum sensors could provide a novel tool to monitor the concentration of contaminants (e.g., heavy metals, paramagnetic ions, etc.) directly in aqueous environments in real time. We note that the spin-pairs in BNNTs are intrinsically limited for electron spin resonance experiments, since their g-factor of $\sim 2.0$ coincides with that of many paramagnetic species, impeding signal discrimination.

Last but not least, BNNT spin pairs hold strong potential for room-temperature bulk nuclear spin hyperpolarization applications, offering a pathway to overcome the main limitations that NV centers in diamond have so far faced in this field[26,28,107,108]. Their hollow geometry and large accessible surface area make BNNTs uniquely suited for scalable polarization transfer: when assembled into densely packed meshes, they can function as flow-through hyperpolarizer devices, ensuring intimate and repeated contact between optically polarized electronic spins and the sample nuclei of surrounding liquids or solids. In contrast to conventional DNP systems that typically require cryogenic temperatures, BNNT-based hyperpolarizers could, in principle, operate entirely under ambient conditions, paving the way for compact, portable, and cost-effective devices.

Before these applications can be realized, several key advances are required, including the deterministic creation of homogeneous defect ensembles to reduce heterogeneity and enhance fluorescence contrast[74,77], as well as improvements in spin coherence, which may be

achievable by operating at higher magnetic fields (e.g., >0.5 T) as recently predicted for hBN defects[109–111].

In conclusion, we present a fundamentally new platform—boron nitride nanotubes (BNNTs)—for spin defect-based quantum sensing. This system overcomes the constraints of traditional bulk and planar quantum materials by introducing a nanoporous, high-surface-area architecture that enhances analyte interactions while preserving robust room-temperature quantum control. At the heart of this system are optically active spin-½ defects, which we attribute to spin-pair $C_?$- defects, previously reported in hBN flakes[71,73,74]. We demonstrated coherent control of randomly oriented nanotube ensembles, achieved a two-order-of-magnitude increase in coherence time via dynamical decoupling protocols, and demonstrated their ability to detect oscillating magnetic fields in the RF frequency range.

In addition, the platform's potential is best exemplified through its integration with microfluidics, where the BNNTs' nanoporous geometry enables efficient analyte interfacing, allowing us to detect paramagnetic $Gd^{3+}$ ions down to low micromolar concentrations. This corresponds to a roughly 1000-fold lower detectable concentration compared to hBN-based sensors[75,76], highlighting the critical role of sensor architecture and surface accessibility in enabling effective detection of paramagnetic species.

Spin-pair defects in BNNTs represent a promising platform for spin-based chemical sensing, with potential applications ranging from nanoscale sensors inside cells and electrochemical devices to local sensors in nanopores and bulk hyperpolarization for enhanced sensitivity in magnetic resonance.

## Methods

### Experimental setup

Initialization of the spin defect ensemble is realized with a 520 nm laser (Cobolt, 06-01) at a power of approximately 150 mW (continuous wave). The excitation laser light is reflected by a dichroic mirror (DMLP550, Thorlabs), after which it is focused on the BNNTs sample by an objective (CFI Plan Apochromat VC 20×, NIKON) with a numerical aperture (NA) of 0.75 and generating a laser spot size of around 20 μm diameter. For the paramagnetic ion sensing experiment, a larger working distance (5 mm) objective was used (100X Nikon CFI60 TU Plan Epi ELWD Infinity). Photoluminescence (PL) is collected by the same objective and focused by a tube lens on either: (1) an avalanche photodiode (APD) (A-Cube-S3000-03, Laser Components) for the spectroscopic path, or (2) a camera (a2A3840-45ucBAS, Basler) for imaging the sample, or (3) a spectrometer (QE Pro, Ocean Optics) for PL emission measurement. The excitation green light and unwanted fluorescence from other defects are filtered out using a long-pass filter with a cut-on wavelength of 550 nm (FEL0550, Thorlabs). The output voltage of the APD is digitized using a data acquisition unit (USB-6221 DAQ, National Instruments). An arbitrary waveform generator (AWG) with up to 2.5 GS/s (AT-AWG-GS2500, Active Technology) is used to synchronize the experiment and generate MW pulse sequences at 250 MHz pulses for spin control, which are mixed with a signal of a local oscillator generator (SG384, Stanford Research Systems) and an IQ mixer (MMIQ0218LXPC 2030, Marki) to reach the final frequency (~2 GHz). The microwave signal is pre-amplified (Mini-Circuits ZX60-153LN-S+) and then passed through a high-power amplifier (ZHL-100W-242+, Mini-Circuits). MW delivery to the BNNTs sample is achieved via a gold strip line, or via a copper strip line in the case of paramagnetic ion sensing. The copper strip line was specifically designed to interface easily with a microfluidic chip, featuring an uninterrupted substrate without cutouts and a mildly textured conductor surface that promotes strong adhesion of the BNNTs, ensuring they remain anchored even under liquid flow. A permanent magnet beneath the sample holder provides a magnetic field in the range of ~50–200 mT. RF signals for AC magnetometry are generated by a waveform generator (DG1022Z, Rigol) connected to a power amplifier

(N-DP340, PRANA). A small wire loop was placed near the sample to deliver RF signals.

### MW delivery setup for characterization and RF sensing

MW strip lines (as shown in Fig. 1e) were designed and fabricated in-house via optical lithography and e-beam Au evaporation to provide microwave fields for coherent electron spin manipulation. Either linear strip lines or omega-shaped strip lines were realized. The microstrip layouts were optimized to maintain impedance matching, with a characteristic impedance of ~50 Ω. The linear strip lines were characterized by a length of 4 mm, and width of 350 μm, while the omega-shaped strip lines were characterized by a ring with inner radius of 500 μm and outer radius of 650 μm (width 150 μm); the ring is interrupted by a 400 μm spacing connected to linear legs tangential to the ring with length of 2 mm. For the realization of the strip lines, optical lithography with a maskless aligner by Heidelberg Instruments was employed, and 10 nm of Ti + 200 nm of Au were evaporated with a Lesker e-beam evaporator on a one-sided-polished 350-μm-thick sapphire substrate to enable thermal dissipation. The chip was glued by thermally conductive paint on a ~1-cm-thick copper block, chosen for its high thermal conductivity. A printed circuit board (PCB) with outer dimensions of 5 cm × 4 cm and an inner hole with a diameter of about 1.4 cm was mounted on the copper block, with the chip being centered on the hole, and the strip line was Au-bonded to the PCB.

### MW delivery setup for paramagnetic ions sensing

A MW strip line (shown in Fig. 4a) was designed in-house and fabricated (CERcuits, Geel, Belgium) to provide MW fields for coherent electron spin manipulation. The microstrip structure was implemented on a printed circuit board (PCB) with outer dimensions of 50.6 mm × 40.6 mm. The substrate material was alumina ($Al_2O_3$) with a thickness of 500 μm, chosen for its high thermal conductivity. The conductive layer and the ground plane were both composed of copper with a thickness of 70 μm. The signal conductor was realized as a straight line with a width of 480 μm, resulting in a characteristic impedance of ~50 Ω. To enhance the local MW field strength in the region of the sample, a ~2 mm-long constriction was introduced at the center of the line, reducing the width to 240 μm. This narrowed region serves to locally increase the current density and thereby intensify the MW magnetic field in the vicinity of the sample positioned above it. The microstrip layout was optimized to maintain impedance matching and minimize signal reflection while providing sufficient field strength for fast coherent spin-state manipulation.

### Sample preparation

The BNNTs used in this study were purchased from NanoIntegris Inc. They are multiwalled and have an average diameter of approximately 50–100 nm. According to the manufacturer, the material has a purity greater than 90%. In addition to boron and nitrogen, elemental analysis indicates the presence of carbon (4.55%), oxygen (5.12%), and magnesium (0.35%). Based on our SEM characterization, the sample is dominated by BNNTs, while a small fraction (estimated ~5–10%) of irregular debris may correspond to residual hBN particles. Throughout this study, the spin defects utilized were naturally present within the sample, which was used without further modification. The BNNTs were transferred onto the devices for MW delivery by dispersing them in ethanol and drop casting them onto the strip line using a pipette, followed by solvent evaporation. Current efforts are being addressed to better characterize the system, especially in terms of defects' spatial distribution and density.

### Spin defects characterization

**ODMR measurements.** The spin defects were optically excited using a 5 μs-long laser pulse, and the electron spin resonance (ESR) transition was driven using a 5 μs MW pulse at ~5 mW power. Fluorescence

**Table 1 | Summary of fitted $T_2$ values, stretch exponents $c$, fit quality (R2), and number of averages ($N_{avg}$) for dynamical decoupling sequences with varying numbers of π-pulses N**

| N (π-pulses) | $T_2$ (ns) ± SD | Stretch exponent $c$ ± SD | R2 | $N_{avg}$ (×$10^3$) |
| --- | --- | --- | --- | --- |
| 1 | 45 ± 5 | 0.90 ± 0.13 | 0.862 | 40 |
| 2 | 68 ± 5 | 1.11 ± 0.15 | 0.908 | 40 |
| 4 | 136 ± 3 | 1.93 ± 0.15 | 0.973 | 40 |
| 8 | 227 ± 2 | 2.09 ± 0.08 | 0.993 | 280 |
| 16 | 384 ± 4 | 2.07 ± 0.09 | 0.991 | 840 |
| 32 | 644 ± 10 | 1.93 ± 0.10 | 0.982 | 600 |
| 64 | 1069 ± 23 | 1.87 ± 0.11 | 0.979 | 560 |
| 128 | 2032 ± 48 | 2.39 ± 0.17 | 0.974 | 2000 |
| 256 | 3391 ± 35 | 2.40 (fixed) | 0.870 | 2000 |
| 512 | 5885 ± 106 | 2.40 (fixed) | 0.810 | 3480 |
| 1024 | 11871 ± 183 | 2.40 (fixed) | 0.723 | 3660 |

contrast was monitored by comparing the signal sequence with a reference sequence (MW off), applied immediately afterward for normalization and noise suppression[112]. Each spectrum in Fig. 2b was recorded at different magnetic field strengths by adjusting the distance of a permanent magnet. The measurement sequence consisted of a 100 μs laser pulse followed by a 10 μs MW pulse. Each spectrum was acquired by dividing the measurement ($R_1$) and reference ($R_2$) readouts as $R_1/R_2$ and recording 10,000 averages per frequency point. In the spectrum of Fig. 2c, a total of 100,000 averages per frequency point and 26 averages of the full sweep were recorded. The spectrum was fitted with a Lorentzian function: $L(f) = \frac{A}{1 + \left(\frac{f - f_0}{\Delta f}\right)^2}$, where $A$ is the amplitude, $f_0$ the resonance frequency, and $\Delta f$ the half-width at half-maximum (HWHM). The fitted parameters are: $A = 5.83 \times 10^{-4}$; $f_0 = 2.19$ GHz, and HWHM = 12.9 MHz.

**Rabi experiments.** Rabi oscillations were recorded by setting the MW frequency to the center of the ODMR spectrum (2.274 GHz) and incrementally sweeping the MW pulse duration. The amplitude of the local oscillator was varied between datasets to generate Rabi oscillations at different MW powers. Considering the power from the preamplifier (18.7 dBm) and the final outpower of the amplifier (50 dBm), the total applied MW power across datasets ranged approximately from 2 W to 40 W, as shown in Fig. 2d. Each data point was obtained by dividing the measurement ($R_1$) and reference ($R_2$) readouts as $R_1/R_2$ using the same normalization and noise cancellation protocol previously described and represents an average over 410,000 repetitions, The experimental Rabi traces were Fourier transformed to (1) extract the precise Rabi frequencies and (2) reveal the presence of a characteristic beating pattern. To examine the dependence of the Rabi frequency on MW power, we extracted the Rabi frequencies from the Fourier transforms of the oscillations (using the main peak) and plotted them as a function of the square root of the MW power. A linear fit of the form $f(x) = ax$ yielded $a = 9.2$ MHz/$\sqrt{W}$ (Fig. 2e).

**$T_1$ measurement.** The longitudinal relaxation time $T_1$ (see Fig. 3a) was measured by varying the delay time T between the initialization and readout laser pulses over a range from 2 μs to ~180 μs and monitoring the fluorescence at each time point (readout $R_1$, see Fig. 3a, inset). To improve signal stability and suppress noise, a reference sequence was applied immediately after each measurement sequence, differing only by the inclusion of a MW π-pulse after the first laser pulse ($R_2$, see Fig. 3a, inset). Each data point was obtained by dividing the fluorescence readouts of the measurement and reference sequences as ($R_1 - R_2$)/($R_1 + R_2$). Every point was averaged 10,000 times, and the full

time-sweep was repeated and averaged over 50 iterations. The resulting decay curve was fitted using a stretched-exponential function of the form: $f(t) = \exp\left[-(t/T_1)^c\right]$. The fit yielded $T_1 = 25.87 \pm 1.03$ μs and $c = 0.665 \pm 0.008$.

**$T_{1\rho}$ measurement.** The rotating frame spin-lattice relaxation time $T_{1\rho}$ was measured using a pulse sequence $[(\pi/2)_x - d - (\text{spinlock})_y - d - (\pi/2)_x]$ and monitoring the resulting fluorescence ($R_1$, see Fig. 3a, inset) while applying step-by-step increments of the spinlock pulse duration. 5 ns-long $\pi/2$-pulses were used, and the delay times $d$ were kept to a value as short as possible (~1 – 2 ns). The amplitude of the spin-lock pulse was set to approximately 30% of the MW amplitude used for the $\pi/2$ pulses, and the MW frequency was detuned by 40 MHz from the exact resonance, as these conditions gave the best signal contrast in our experiments. To enable noise cancellation, a referencing scheme was employed by alternating the final $(\pi/2)_x$ pulse with a $(3\pi/2)_x$ pulse in successive measurements ($R_2$, see Fig. 3a, inset). Each data point was obtained by computing ($R_1$-$R_2$)/($R_1 + R_2$) and averaged over 100,000 repetitions to improve the signal-to-noise ratio. The resulting curve fits a stretched exponential decay of the form: $f(t) = a \cdot \exp\left[-(t/T_{1\rho})^c\right]$. The fit yielded $T_{1\rho} = 16.57 \pm 1.66$ μs and a stretch factor $c = 0.640 \pm 0.036$, with $a = 1$ fixed.

**$T_2$ measurements.** The spin-echo sequence followed the standard protocol $[(\pi/2)_y - \tau - (\pi)_x - \tau - (\pi/2)_y]$, where the interpulse delay τ was swept from 10 ns to ~200 ns. To enable noise cancellation, a referencing scheme was employed by alternating the final $(\pi/2)_y$ pulse with a $(3\pi/2)_y$ pulse in successive measurements. Each data point was obtained by computing ($R_1 - R_2$)/($R_1 + R_2$) and averaging 10,000 times, and the full sweep over τ values was repeated 100 times for improved signal-to-noise. The resulting decay curve was fitted using a stretched exponential function of the form: $f(t) = a \cdot \exp\left[-(t/T_2)^c\right]$, with $a = 1$ fixed. The fit yielded a coherence time $T_2 = 44.71 \pm 4.7$ ns and a stretch factor c = 0.902 ± 0.133.

**CPMG measurements.** We employed the same general approach used for the $T_2$ measurement—monitoring the spin-echo signal while increasing the free evolution time τ. Multiple decoherence curves were acquired for pulse sequences containing an increasing number $N$ of π-pulses, up to $N = 1000$. To enable direct comparison across datasets, a consistent normalization procedure was applied. First, the time axis of each dataset was rescaled by a factor $S = 2N$, corresponding to the total evolution time $T$ between the initial and final π/2 pulses in the respective sequences. Next, the decay curve from the standard spin-echo experiment (i.e., $N = 1$) was fitted with the function: $f(T) = A \cdot \exp\left[-(T/T_2)^c\right]$. The amplitude $A$ yields the spin-echo contrast corresponding to $T = 0$. All datasets were normalized to this value, and fitted using the same function, with the amplitudes kept to $A = 1$ (Table 1). The dependence of the coherence time $T_2$ on the number of π-pulses, as shown in the inset of Fig. 2b, was fitted with a power-law function of the form $f(N) = a \cdot N^S$, yielding fit parameters $a = (33 \pm 2)$ ns and s = 0.79 ± 0.01. The error bars correspond to ± 1.96 $\sigma_{res}$, where $\sigma_{res}$ ($\approx \pm 1.35 \times 10^{-7}$) is the residual standard deviation of the global power-law fit, applied uniformly to all points.

**Calibration of the MW pulses.** Optimal conditions for CPMG and spin-lock experiments were not achieved by simply driving the spins precisely on resonance or by selecting π/2 and π pulse durations based on Rabi revivals or their Fourier transforms. Instead, we found that more reliable results were obtained by fixing the pulse durations (e.g., 5 ns and 10 ns for the π/2 and π pulses, respectively) and empirically tuning the local oscillator amplitude to maximize the spin echo signal. In the case of spin-lock experiments, we consistently observed enhanced signal contrast when the microwave drive was deliberately set slightly off-resonance.

**RF sensing.** We applied a pulse sequence synchronized with the sensing RF signal, consisting of concatenated 2-π-pulse spin echo (XY-2) subsequences. These subsequences were repeated such that the timing between them matched an integer multiple of the RF period. Each sequence included 5 μs laser pulses for initialization and readout, and 5 and 10 ns as π/2- and π-pulses, respectively, for MW manipulation. The inter π-pulse delay was set to $\tau = 16$ ns corresponding to a target RF frequency of $\nu_{RF} = 15.625$ MHz. For generating a relative frequency offset $\Delta\nu = 1000$ Hz, the RF frequency was shifted slightly to $\nu_{RF} = 15.626$ MHz. The total sampling time was $t_s = 2$ s. The resulting time-domain signal (shown in Fig. 4b, c) was Fourier transformed, and the absolute value was plotted in Fig. 4d, revealing a narrow peak with a full width at half maximum of approximately 1 Hz. The final signal was obtained after averaging 1000 repetitions. Using the calibration procedure described in Ref. 53, we estimated a magnetic field sensitivity of ~ 20 μT/√Hz.

**Paramagnetic ion sensing.** A small drop of the BNNT dispersion—prepared as described in the Sample Preparation section—was drop-cast onto the microwave strip line and allowed to dry completely as the solvent evaporated. This strip line was then glued to a custom-made glass microcuvette, designed in-house to enable continuous liquid flow through the channel (Fig. 4e). ODMR and $T_1$ relaxation measurements were performed using the protocols described above, with the BNNT mesh sensor in contact with either pure water or $Gd^{3+}$ ion solutions at increasing concentrations, ranging from 1 μM to 100 mM. Between measurement sessions, the sensor was reused following a cleaning procedure that involved flushing with several milliliters of distilled water, an acidic wash using $H_2SO_4$ at pH 2, and a final rinse with pure distilled water. For the ODMR data shown in Fig. 4f, $T_1$ measurements were obtained using 20,000 averages and 100 sweeps for the $Gd^{3+}$ solution, and 20,000 averages with 20 sweeps for pure water. The corresponding ODMR spectra in the inset were acquired by averaging 20,000 repetitions per point and performing three full spectral sweeps for pure water, and 100 sweeps for the 1 mM $Gd^{3+}$ solution. In the data presented in Fig. 4g, the ODMR signal was monitored while $Gd^{3+}$ solutions of increasing concentration (from 1 μM to 100 mM) were injected into the microchannel. At each concentration step, the channel was first flushed with 1 mL of the respective solution to allow the sensor to equilibrate. Each ODMR spectrum was then recorded using 20,000 averages and 9 sweeps (~ 4 min of acquisition time per spectrum), except for the last three concentrations (from 1 mM to 100 mM), for which 48 sweeps (~ 20 min of acquisition time) were used to improve signal quality. After fitting each spectrum with a Lorentzian function, the maximum ODMR contrast was plotted as a function of $Gd^{3+}$ concentration to extract the concentration dependence. The data were well described by a generalized Langmuir (or Hill-like) function of the form $C_0/(1 + ([Gd^{3+}]/K_d)^n) + A$, where $C_0$ is the unperturbed ODMR contrast (distilled water), $K_d$ is the dissociation constant, $n$ is the Hill coefficient, and $A$ is a saturation offset accounting for the nonzero baseline contrast observed at high $Gd^{3+}$ concentrations. The best-fit parameters were: $C_0 = 0.1195 \pm 0.0045$,    $K_d = 1.69 \times 10^{-5} \pm 4.20 \times 10^{-6}$,    $n = 0.78 \pm 0.11$, $A = 0.0272 \pm 0.0020$.

## Data availability

Source data are provided with this paper. The Source Data used in this study are available in the ZENODO database under accession code [https://doi.org/10.5281/zenodo.17700056]. All other data that support the findings of this study are available from the corresponding author upon reasonable request. Source Data are provided with this paper.

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

## Acknowledgments

This project has been funded by the Bayerisches Staatministerium für Wissenschaft und Kunst through project IQSense via the Munich Quantum Valley (MQV), the Deutsche Forschungsgemeinschaft (DFG, German Research Foundation)—412351169 within the Emmy Noether program and the European Research Council (ERC) under the European Union's Horizon 2020 research and innovation program (Grant Agreement No. 948049). The authors acknowledge support and seed funding by the DFG under Germany's Excellence Strategy–EXC 2089/1-390776260 and the EXC-2111 390814868.

## Author contributions

R.R. and D.B.B. conceived the idea and designed the research. R.R. carried out the experiments and simulations. R.R. and A.A.H. built the experimental setup, with L.N. contributing to its optimization. E.B. provided the microstructure for microwave delivery used in the characterization, coherence extension, and RF sensing studies. N.vG. designed and supplied the structure for microwave delivery used in the ion sensing experiments. L.N. carried out the SEM characterization of the BNNTs. R.R. and L.N. implemented the integration of the BNNTs sensor with microfluidics. J.J.F. provided guidance on theoretical and experimental aspects. R.R. and D.B.B. analyzed the data. R.R. and D.B.B. wrote the manuscript with input from all authors.

## Funding

## Competing interests

All authors declare no competing interests.
