## [Peer Review File · Nature Communications]

Quantum sensing with spin defects in boron nitride nanotubes

Corresponding Author: Dr Roberto Rizzato

Version 1:

Reviewer comments:

Reviewer #1

(Remarks to the Author)

The authors have thoroughly addressed my comments and made substantial improvements to the manuscript. The figure 1 has been significantly enhanced for clarity and presentation quality. In my view, the revised version meets the high standards of Nature Communications, and I believe the manuscript should be accepted for publication.

Reviewer #2

(Remarks to the Author)

The authors have addressed my comments and provided extended explanations to rationalize the conclusions of the manuscript. I recommend it for publication in Nature Communications. I also found authors' response on the unique architecture/high surface accessibility of BNNT and underlying mechanisms for the improved performance of BNNT (compared with the flat hBN case) is intriguing. I think adding some of the discussions in the paper could help readers to further understand the conclusions.

Reviewer #3

(Remarks to the Author)

The authors have made extensive revisions addressing all my and the other referee's comments. As I stated in my previous report, this is a very good paper that deserves to be published in a good journal. I think Nat Comms is a great fit and I'm happy to recommend publication as is. I congratulate the authors on the great work, and I look forward to seeing what the BNNT platform can enable in the future.

Reviewer #4

(Remarks to the Author)

Point by Point letter

REVIEWERS' COMMENTS

Reviewer #1 (Remarks to the Author): The authors have thoroughly addressed my comments and made substantial improvements to the manuscript. The figure 1 has been significantly enhanced for clarity and presentation quality. In my view, the revised version meets the high standards of Nature Communications, and I believe the manuscript should be accepted for publication.

Reviewer #2 (Remarks to the Author): The authors have addressed my comments and provided extended explanations to rationalize the conclusions of the manuscript. I recommend it for publication in Nature Communications. I also found authors' response on the unique architecture/high surface accessibility of BNNT and underlying mechanisms for the improved performance of BNNT (compared with the flat hBN case) is intriguing. I think adding some of the discussions in the paper could help readers to further understand the conclusions.

We thank the Reviewer for their efforts in helping improve our manuscript. We believe our working hypothesis is now clearly stated. This appears at the end of the section on enhanced sensing of paramagnetic ions, where we added: *“Our working hypothesis is that the improved performance of our BNNT sensor is driven by its unique architecture and high surface accessibility, rather than the intrinsic qualities of the spin defects themselves. Unlike flat hBN flakes, the BNNT mesh forms a nanoporous, high-surface-area network that may enable local confinement of analytes in close proximity to spin-active regions. The nanotubes are hollow and open at both ends, allowing analytes to diffuse into the inner cavity and interact with spin defects within the tube walls. In addition, the cylindrical geometry provides radial access, in contrast to the one-sided exposure in flat 2D materials and may facilitate local concentration effects through analyte accumulation within the mesh.”*

Reviewer #3 (Remarks to the Author): The authors have made extensive revisions addressing all my and the other referee's comments. As I stated in my previous report, this is a very good paper that deserves to be published in a good journal. I think Nat Comms is a great fit and I'm happy to recommend publication as is. I congratulate the authors on the great work, and I look forward to seeing what the BNNT platform can enable in the future.

Reviewer #4 (Remarks to the Author): I co-reviewed this manuscript with one of the reviewers who provided the listed reports. This is part of the Nature Communications initiative to facilitate training in peer review and to provide appropriate recognition for Early Career Researchers who co-review manuscripts.